# THE UNCANNY SIMILARITY OF RECURRENCE AND DEPTH

**Avi Schwarzschild**[*]
Department of Mathematics
University of Maryland
College Park, MD, USA
`avi1@umd.edu`

**Arjun Gupta**[*]
Department of Robotics
University of Maryland
College Park, MD, USA
`arjung15@umd.edu`

**Amin Ghiasi**
Department of Computer Science
University of Maryland
College Park, MD, USA

**Micah Goldblum**
Department of Computer Science
University of Maryland
College Park, MD, USA

**Tom Goldstein**
Department of Computer Science
University of Maryland
College Park, MD, USA

## ABSTRACT

It is widely believed that deep neural networks contain layer specialization, wherein neural networks extract hierarchical features representing edges and patterns in shallow layers and complete objects in deeper layers. Unlike common feed-forward models that have distinct filters at each layer, recurrent networks reuse the same parameters at various depths. In this work, we observe that recurrent models exhibit the same hierarchical behaviors and the same performance benefits with depth as feed-forward networks despite reusing the same filters at every recurrence. By training models of various feed-forward and recurrent architectures on several datasets for image classification as well as maze solving, we show that recurrent networks have the ability to closely emulate the behavior of non-recurrent deep models, often doing so with far fewer parameters.

## 1 INTRODUCTION

It is well-known that adding depth to neural architectures can often enhance performance on hard problems (He et al., 2016; Huang et al., 2017). State-of-the-art models contain hundreds of distinct layers (Tan & Le, 2019; Brock et al., 2021). However, it is not obvious that recurrent networks can experience performance boosts by conducting additional iterations, since this does not add any parameters. On the other hand, increasing the number of iterations allows the network to engage in additional processing. In this work, we refer to the number of sequential layers (possible not distinct) as effective depth. While it might seem that the addition of new parameters is a key factor in the behavior of deep architectures, we show that recurrent networks can exhibit improved behavior, closely mirroring that of deep feed-forward models, simply by adding recurrence iterations (to increase the effective depth) and no new parameters at all.

In addition to the success of very deep networks, a number of works propose plausible concrete benefits of deep models. In the generalization literature, the sharp-flat hypothesis submits that networks with more parameters, often those which are very wide or deep, are biased towards flat minima of the loss landscape which are thought to coincide with better generalization (Keskar et al., 2016; Huang et al., 2020; Foret et al., 2020). Feed-forward networks are also widely believed to have *layer specialization*. That is, the layers in a feed-forward network are thought to have distinct convolutional filters that build on top of one another to sequentially extract hierarchical features (Olah et al., 2017; Geirhos et al., 2018). For example, the filters in shallow layers of image classifiers may be finely tuned to detect edges and textures while later filters may be precisely tailored for detecting semantic structures such as noses and eyes (Yosinski et al., 2015). In contrast, the filters in early and deep iterations of recurrent networks have the very same parameters. Despite their lack of layer-wise

---

[*]Equal contribution.

specialization and despite the fact that increasing the number of recurrences does not increase the parameter count, we find that recurrent networks can emulate both the generalization performance and the hierarchical structure of deep feed-forward networks, and the relationship between depth and performance is similar regardless of whether recurrence or distinct layers are used.

Our contributions can be summarized as follows:

- We show that image classification accuracy changes with depth in remarkably similar ways for both recurrent and feed-forward networks.
- We introduce datasets of mazes, a sequential reasoning task, on which we show the same boosts in performance by adding depth to recurrent and to feed-forward models.
- With optimization based feature visualizations, we show that recurrent and feed-forward networks extract the very same types of features at different depths.

## 1.1 RELATED WORKS

Recurrent networks are typically used to handle sequential inputs of arbitrary length, for example in text classification, stock price prediction, and protein structure prediction (Lai et al., 2015; Borovkova & Tsiamas, 2019; Goldblum et al., 2020; Baldi & Pollastri, 2003). We instead study the use of recurrent layers in place of sequential layers in convolutional networks and on non-sequential inputs, such as images. Prior work on recurrent layers, or equivalently depth-wise weight sharing, also includes studies on image segmentation and classification as well as sequence-based tasks. For example, Pinheiro & Collobert (2014) develop fast and parameter-efficient methods for segmentation using recurrent convolutional layers. Alom et al. (2018) use similar techniques for medical imaging. Also, recurrent layers have been used to improve performance on image classification benchmarks (Liang & Hu, 2015). Recent work developing transformer models for image classification shows that weight sharing can reduce the size of otherwise large models without incurring a large performance decrease (Jaegle et al., 2021).

Additionally, the architectures we study are related to weight tying. Eigen et al. (2013) explore very similar dynamics in CNNs, but they do not address MLPs or ResNets, and Boulch (2017) study weight sharing only in ResNets. More importantly, neither of those works have analysis beyond accuracy metrics, whereas we delve into what exactly is happening at various depths with feature visiualization, linear separability probes, and feature space similarity metrics. Bai et al. (2018) propose similar weight tying for sequences, and they further extend this work to include equilibrium models, where infinite depth networks are found with root-finding techniques (Bai et al., 2018; 2019). Equilibrium models and neural ODEs make use of the fact that hidden states in a recurrent network can converge to a fixed point, training models with differential equation solvers or root finding algorithms (Chen et al., 2018; Bai et al., 2019).

Prior work towards understanding the relationship between residual and highway networks and recurrent architectures reveals phenomena related to our study. Greff et al. (2016) present a view of highway networks wherein they iteratively refine representations. The recurrent structures we study could easily be thought of in the same way, however they are not addressed in that work. Also, Jastrzebski et al. (2017) primarily focus on deepening the understanding of iterative refinement in ResNets. While our results may strengthen the argument that ResNets are naturally doing something iterative, this is tangential to our broader claims that for several families of models depth can be achieved with distinct layers or with recurrent ones. Similarly, Liao & Poggio (2016) show that deep RNNs can be rephrased as ResNets with weight sharing and even include a study on batch normalization in recurrent layers. Our work builds on theirs to further elucidate the similarity of depth and recurrence, specifically, we carry out quantitative and qualitative comparisons between the deep features, as well as performance, of recurrent and analogous feed-forward architectures as their depth scales. We analyze a variety of additional models, including multi-layer perceptrons and convolutional architectures which do not include residual connections.

Our work expands the scope of the aforementioned studies in several key ways. First, we do not focus on model compression, but we instead analyze the relationship between recurrent networks and models whose layers contain distinct parameters. Second, we use a standard neural network training process, and we study a variety of architectures. Finally, we conduct our experiments on image data, as well as a reasoning task, rather than sequences, so that our recurrent models can be viewed

as standard fully-connected or residual networks but with additional weight-sharing constraints. For each family of architectures and each dataset we study, we observe various trends as depth, or equivalently the number of recurrence iterations, is increased that are consistent from recurrent to feed-forward models. These relationships are often not simple, for example classification accuracy does not vary monotonically with depth. However, the trends are consistent whether depth is added via distinct layers or with iterations. We are the first to our knowledge to make qualitative and quantitative observations that recurrence mimics depth.

## 2 RECURRENT ARCHITECTURES

Widely used architectures, such as ResNet and DenseNet, scale up depth in order to boost performance on benchmark image classification tasks (He et al., 2016; Huang et al., 2017). We show that this trend also holds true in networks where recurrence replaces depth. Typically, models are made deeper by adding layers with new parameters, but recurrent networks can be made deeper by recurring through modules more times without adding any parameters.

To discuss this further, it is useful to define *effective depth*. Let a layer of a neural network be denoted by $\ell$. Then, a $p$-layer feed-forward network can be defined by a function $f$ that maps an input $x$ to a label $y$ where

$$f(x) = \ell_p \circ \ell_{p-1} \circ \cdots \circ \ell_2 \circ \ell_1(x).$$

In this general formulation, each member of the set $\{\ell_i\}_{i=1}^{p}$ can be a different function, including compositions of linear and nonlinear functions. The recurrent networks in this study are of the following form.

$$f(x) = \ell_p \circ \cdots \circ \ell_{k+1} \circ m_r \circ m_r \circ \cdots \circ \ell_k \circ \cdots \circ \ell_1(x) \tag{1}$$

We let $m_r(x) = \ell_q^{(m)} \circ \cdots \circ \ell_1^{(m)}(x)$ and call it a $q$-layer *recurrent module*, which is applied successively between the first and last layers. The *effective depth* of a network (feed-forward or recurrent) is the number of layers, not necessarily unique, composed with one another to form $f$. Thus, the network in Equation equation 1 has an effective depth of $p + nq$, where $n$ is the number of iterations of the recurrent module. A more concrete example is a feed-forward network with seven layers, which has the same effective depth as a recurrent model with two layers before the recurrent module, two layers after the recurrent module, and a single-layer recurrent module that runs for three iterations. See Figure 1 for a graphical representation the effective depth measurement.

We train and test models of varying effective depths on ImageNet, CIFAR-10, EMNIST, and SVHN to study the relationship between depth and recurrence (Russakovsky et al., 2015; Krizhevsky, 2009; Cohen et al., 2017; Netzer et al., 2011). For every architecture style and every dataset, we see that depth can be emulated by recurrence. The details of the particular recurrent and feed-forward models we use in these experiments are outlined below. For specific training hyperparameters for every experiment, see Appendix A.2.[1]

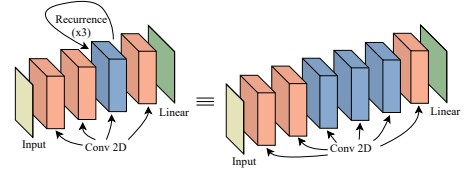

Figure 1: Effective depth measures the depth of the unrolled network. For example, a network with 3 iterations of its single-layer recurrent module and 4 additional layers has an effective depth of 7.

The specific architectures we study in the context of image classification include families of multi-layer perceptrons (MLPs), ConvNets, and residual networks. Each recurrent model contains an initial set of layers, which project into feature space, followed by a module whose inputs and outputs have the same shape. As a result of the latter property, this module can be iteratively applied in a recurrent fashion an arbitrary number of times. The feed-forward analogues of these recurrent models instead stack several unique instances of the internal module on top of each other. In other words, our recurrent models can be considered the same as their feed-forward counterparts but with depth-wise weight sharing between each instance of the internal module.

---

[1]Code for reproducing the experiments in this paper is available in the code repository at https://github.com/Arjung27/DeepThinking.

## 2.1 MULTI-LAYER PERCEPTRONS

The MLPs we study are defined by the width of their internal modules which is specified per dataset in Appendix A.1. We examine feed-forward and recurrent MLPs with effective depths from 3 to 10. The recurrent models have non-recurrent layers at the beginning and end, and they have one layer in between that can be recurred. For example, if a recurrent MLP has effective depth five, then the model is trained and tested with three iterations of this single recurrent layer. A feed-forward MLP of depth five will have the exact same architecture, but instead with three distinct fully-connected layers between the first and last layers. All MLPs we use have ReLU activation functions between full-connected layers.

## 2.2 CONVOLUTIONAL NETWORKS

Similarly, we study a group of ConvNets which have two convolutional layers before the internal module and one convolutional and one fully connected layer afterward. The internal module has a single convolutional layer with a ReLU activation, which can be repeated any number of times to achieve the desired effective depth. Here too, the output dimension of the second layer defines the width of the model, and this is indicated in Appendix A.1 for specific models. Each convolutional layer is followed by a ReLU activation, and there are maximum pooling layers before and after the last convolutional layer. We train and test feed-forward and recurrent ConvNets with effective depths from 4 to 8.

## 2.3 RESIDUAL NETWORKS

We also employ networks with residual blocks like those introduced by He et al. (2016). The internal modules are residual blocks with four convolutional layers, so recurrent models have a single recurrent residual block with four convolutional layers. There is one convolutional layer before the residual block and one after, which is followed by a linear layer. The number of channels in the output of the first convolutional layer defines the width of these models. Each convolutional layer is followed by a ReLU activation. As is common with residual networks, we use average pooling before the linear classifier. The residual networks we study have effective depths from 7 to 19.

## 3 MATCHING IMAGE CLASSIFICATION ACCURACIES

Across all model families, we observe that adding iterations to recurrent models mimics adding layers to their feed-forward counterparts. This result is not intuitive for three reasons. First, the trend in performance as a function of effective depth is not always monotonic and yet the effects of adding iterations to recurrent models closely match the trends when adding unique layers to feed-forward networks. Second, adding layers to feed-forward networks greatly increases the number of parameters, while recurrent models experience these same trends without adding any parameters at all. On the CIFAR-10 dataset, for example, our recurrent residual networks each contain 12M parameters, and yet models with effective depth of 15 perform almost exactly as well as a 15-layer feed-forward residual network with 31M parameters. Third, recurrent models do not have distinct filters that are used in early and deep layers. Therefore, previous intuition that early filters are tuned for edge detection while later filters are tailored to pick up higher order features cannot apply to the recurrent models (Olah et al., 2017; Yosinski et al., 2015).

In Figure 2, the striking result is how closely the blue and orange markers follow each other. For example, on SVHN, we see from the triangular markers that residual networks generally perform better as they get deeper and that recurrent residual models with multiple iterations perform as well as their non-recurrent counterparts. On CIFAR-10, we see that MLP performance as a function of depth is not monotonic, yet we also see the same pattern here as we add iterations to recurrent models as we do when we add distinct layers to non-recurrent networks. The increasing trend in the ImageNet accuracies, which are matched by the recurrent and feed-forward models, suggests that even deeper models might perform even better. In fact, we can train a ResNet-50 and achieve 76.48% accuracy on the testset. While training our recurrent models, which do not have striding or pooling, at that depth is dificult, we can training a ResNet-50 modified so that layers within residual blocks that permit inputs/outputs of the same shape share weights. This model is 75.13% accurate on the testset. These

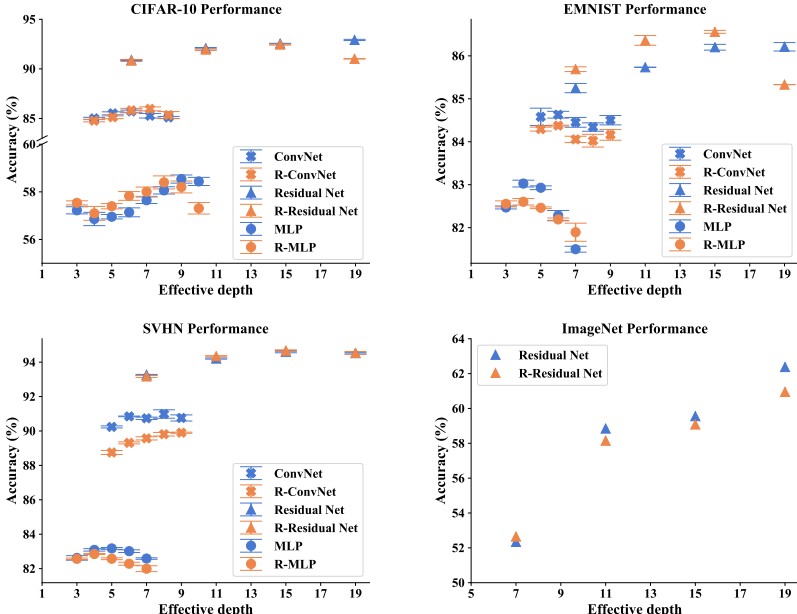

Figure 2: Average accuracy of each model. The x-axis represents effective depth and the y-axis measures test accuracy. The marker shapes indicate architectures, and the blue and orange markers indicate feed-forward and recurrent models, respectively. The horizontal bars are ± one standard error from the averages over several trials. Across all three families of neural networks, recurrence can imitate depth. Note, the $y$-axes are scaled per dataset to best show each of the trends.

experiments show that the effect of added depth in these settings is consistent whether this depth is achieved by iterating a recurrent module or adding distinct layers.

## 3.1 SEPARATE BATCH NORMALIZATION STATISTICS FOR EACH RECURRENCE

The models described above comprise a range of architectures for which recurrence can mimic depth. However, those models do not employ the modern architectural feature batch normalization (Ioffe & Szegedy, 2015). One might wonder if the performance of recurrent networks is boosted even further by the tricks that help state-of-the-art models achieve such high performance. We compare a recurrent residual network to the well-known ResNet-18 architecture (He et al., 2016). There is one complexity that arises when adding batch normalization to recurrent models; despite sharing the same filters, the feature statistics after each iteration of the recurrent module may vary, so we must use a distinct batch normalization layer for each iteration. Although it is not our goal to further the state of the art, Table 1 shows that these recurrent models perform nearly as well as ResNet-18, despite containing only a third of the parameters. This recurrent architecture is not identical to the models used above, rather it is designed to be as similar to ResNet-18 as possible; see Appendix A.1 for more details.

Table 1: **Models with batch normalization.** Performance shown for three different classification datasets. We report averages (± one standard error) from three trials.

| Dataset | Model | Params | Acc (%) |
|---|---|---|---|
| CIFAR-10 | Ours | 3.56M | $93.99 \pm 0.10$ |
| | ResNet-18 | 11.2M | $94.69 \pm 0.03$ |
| SVHN | Ours | 3.56M | $96.04 \pm 0.08$ |
| | ResNet-18 | 11.2M | $95.48 \pm 0.11$ |
| EMNIST | Ours | 3.57M | $87.42 \pm 0.19$ |
| | ResNet-18 | 11.2M | $87.71 \pm 0.09$ |

# 4 THE MAZE PROBLEM

To delve deeper into the capabilities of recurrent networks, we shift our attention from image classification and introduce the task of solving mazes. In this new dataset, the mazes are represented by three-channel images where the permissible regions are white, the walls are black, and the start and end points are green and red squares, respectively. We generate 180,000 mazes in total, 60,000 examples each of small (9x9), medium (11x11), and large mazes (13x13). We split each set into 50,000 training samples and 10,000 testing samples. The solutions are represented as a single-channel output of the same spatial dimensions as the input maze, with 1's on the optimal path from the green square to the red square and 0's elsewhere. For a model to "solve" a maze, it must output a binary segmentation map – each pixel in the input is either on the optimal path or not. We only count a solution as correct if it matches labels every pixel correctly. See Figure 3 for an example maze.

## 4.1 MAZE GENERATION

The mazes are generated using a depth-first search based algorithm (Hill, 2017). We initialize a grid of cells with walls blocking the path from each cell to its neighbors. Then, starting at an arbitrary cell in the maze, we randomly choose one of its unvisited neighbors and remove the walls between the two and move into the neighboring cell. This process continues until we arrive at a cell with no unvisited neighbors, at which point we trace back to the last cell that has unvisited neighbors. This generation algorithm provides mazes with a path from each cell to every other cell in the initialized grid. The images are made by plotting all initial cells and all removed walls as white squares, leaving cell boundaries that were not deleted in black. We

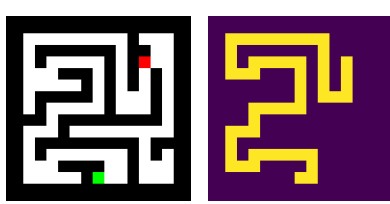

Figure 3: A sample maze and its solution. This is an example of an input/target pair used for training and testing neural maze solvers.

also denote the "start" and "end" points of the maze with green and red squares, respectively. After the mazes are generated, they are solved with Dijkstra's algorithm, and the solutions are saved as described above. Figure 4 shows the distribution of optimal path lengths in each of the sets of mazes. There is a non-zero chance of repeating a maze, however we observe that fewer than 0.5% of the mazes are duplicated within each of the training sets, and the training and testing sets do not overlap.

## 4.2 MAZE SOLVING NETWORKS

We extend our inquiry into recurrence and depth to neural networks that solve mazes. Specifically, we look at residual networks' performance on the maze dataset. The networks take a maze image as input and output a binary classification for each pixel in the image. These models have one convolutional layer before the internal module and three convolutional layers after. Similar to classification networks, the internal modules are residual blocks and a single block is repeated in recurrent models while feed-forward networks have a stack of distinct blocks. We employ residual networks with effective depths from 8 to 44 layers.

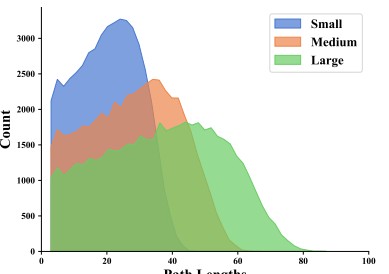

Figure 4: The distribution of optimal path lengths in each dataset (small, medium, and large mazes).

## 4.3 SOLVING
### MAZES REQUIRES DEPTH (OR RECURRENCE)

We train several recurrent and non-recurrent models of effective depths ranging from 8 to 44 on each of the three maze datasets, and we see that deeper models perform better in general. Figure 5 shows these results. More importantly, we observe that recurrent models perform almost exactly as well as feed-forward models with equal effective depth. Another crucial finding confirmed by these plots is that larger mazes are indeed harder to solve. We see that with a fixed effective depth, models perform the best on small mazes, then medium mazes, and the worst on large mazes. This trend matches the

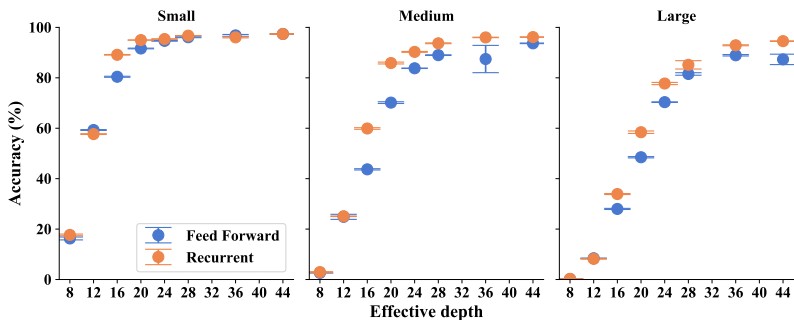

Figure 5: Recurrent models perform similarly to feed-forward models of the same effective depth. All models here are residual models as described in Section 4.

intuition that larger mazes require more computation to solve, and it confirms that these datasets do, in fact, constitute problems of varying complexity.

One candidate explanation for these trends is the *receptive field*, or the range of pixels in the input that determine the output at a given pixel. Intuitively, solving mazes requires global information rather than just local information which suggests that the receptive field of successful models must be large. While the receptive field certainly grows with depth, this does not fully account for the behavior we see. By computing the receptive field for each effective depth, we can determine whether the accuracy improves with even more depth beyond what is needed for a complete receptive field. For the small mazes, the receptive field covers the entire maze after eight $3 \times 3$ convolutions, and for the large mazes, this takes 12 such convolutions. In Figure 5, we see performance rise in all three plots even after effective depth grows beyond 20. This indicates that receptive field does not fully explain the benefits of effective depth – there is further improvement brought about by adding depth, whether by adding unique parameters or reusing filters in the recurrent models.

We can further test this by training identical architectures to those described above, except that the $3 \times 3$ convolutional filters are dilated (this increases the receptive field without adding parameters or depth). In fact, these models are better at solving mazes, but most importantly, we also see exactly the same relationship between recurrence and depth. This confirms that recurrence has a strong benefit beyond merely increasing the receptive field. See Table 2 for numerical results.

Table 2: **Dilated Filters.** The average accuracy of models with dilated filters trained and tested on small mazes.

|              | Effective Depth | | | |
| Model        | 8     | 12    | 16    | 20    |
|--------------|-------|-------|-------|-------|
| Recurrent    | 75.07 | 90.41 | 94.70 | 95.19 |
| Feed-forward | 75.59 | 90.58 | 93.84 | 95.00 |

## 5 RECURRENT MODELS REUSE FEATURES

Test accuracy is only one point of comparison, and the remarkable similarities in Figure 2 raise the question: What are the recurrent models doing? Are they, in fact, recycling filters, or are some filters only active on a specific iteration? In order to answer these questions, we look at the number of positive activations in each channel of the feature map after each recurrence iteration for a batch of 1,000 randomly selected CIFAR-10 test set images. For each image-filter pair, we divide the number of activations at each iteration by the number of activations at the most active iteration (for that pair). If this measurement is positive on iterations other than the maximum, then the filter is being reused on multiple iterations. In Figure 6, it is clear that a large portion of the filters have activation patterns on their least active iteration with at least 20% of the activity of their most active iteration. This observation leads us to conclude that these recurrent networks are indeed reusing filters, further supporting the idea that the very same filters can be used in shallow layers as in deep layers without a loss in performance. See Appendix A.4 for similar figures from different models and datasets.

## 5.1 RECURRENT AND FEED-FORWARD MODELS SIMILARLY DISENTANGLE CLASSES

Table 3: **Classifiability of Features.** CIFAR-10 classification accuracy of linear classifiers trained on intermediate features output by each residual block (or each iteration) in networks with 19-layer effective depths. Each cell reports the percentage of training/testing images correctly labeled.

|  | Block #1 | Block #2 | Block #3 | Block #4 |
|---|---|---|---|---|
| Feed Forward | 84.7 / 81.6 | 94.1 / 86.9 | 99.3 / 89.8 | 100.0 / 90.6 |
| Recurrent | 86.3 / 82.3 | 95.9 / 87.5 | 99.5 / 89.4 | 99.9 / 90.3 |

In order to compare the features that image classifiers extract, we train linear classifiers on intermediate features from various layers of the network. In trying to classify feature maps with linear models, we can analyze a network's progress towards disentangling the features of different classes. For this experiment, we examine the features of CIFAR-10 images output by both feed-forward and recurrent residual networks. Similar to the method used by Kaya et al. (2019), we use the features output by each residual block to train a linear model to predict the labels based on these feature vectors.

Using residual networks with effective depth of 19 (as this is the best performing feed-forward depth), we show that the deeper features output by each residual block, or iteration in the recurrent model, are increasingly separable. More importantly, as shown in Table 3, the feed-forward and recurrent models are separating features very similarly at corresponding points in the networks. Details on the training of these linear classifiers, in addition to results with pooling are in Appendix A.5.

Another tool for quantifying relationships between deep neural networks is feature space similarity metrics. We employ the techniques proposed by Kornblith et al. (2019) to compare models trained on CIFAR-10 and obtain a similarity score between zero and one. When we compare feed-forward ConvNets with their recurrent counterparts, the results indicate that models with the same effective depths are indeed very simlar. Specifically, the features extracted by an eight-layer feed-forward ConvNet are more similar to those extracted by its recurrent counterpart with the same effective depth (CKA = 0.843) than they are to those of feed-forward models of different depths (e.g. CKA = 0.819 for a ConvNet with 4 layers). That is, feed-forward and recurrent networks of the same effective depth are more similar to each other than they are to models of different depths.

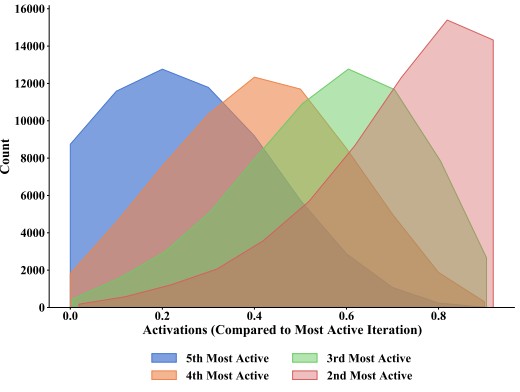

Figure 6: The number of active neurons after each iteration of a recurrent ConvNet with 5 iterations. Note that the most active iteration is left off this plot since we are normalizing by the number activations at that iteration.

## 5.2 VISUALIZING THE ROLES OF RECURRENT AND FEED-FORWARD LAYERS

In previous sections, we performed quantitative analysis of the classification performance of recurrent and feed-forward models. In this section, we examine the qualitative visual behavior that transpires inside these networks. It is widely believed that deep networks contain a hierarchical structure in which early layers specialize in detecting simple structures such as edges and textures, while late layers are tuned for detecting semantically complex structures such as flowers or faces (Yosinski et al., 2015). In contrast, early and late layers of recurrent models share exactly the same parameters and therefore are not tuned specifically for extracting individual simple or complex visual features. Despite recurrent models applying the same filters over and over again, we find that these same filters detect simple features such as edges and textures in early layers and more abstract features in later layers. To this end, we visualize the roles of individual filters by constructing images which maximize a filter's activations at a particular layer or equivalently at a particular recurrence iteration.

| Depth | Recurrent | Feed Forward |
|---|---|---|

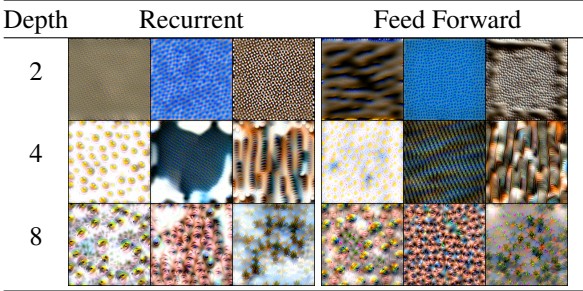
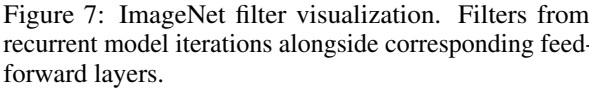

| Depth | Recurrent | Feed Forward |
|---|---|---|

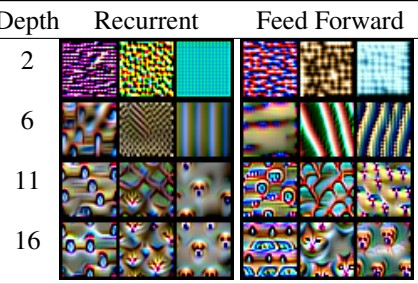

Figure 7: ImageNet filter visualization. Filters from recurrent model iterations alongside corresponding feed-forward layers.

Figure 8: CIFAR-10 filter visualization. Filters from recurrent model iterations and corresponding feed-forward layers.

We follow a similar approach to Yosinski et al. (2015). We also employ a regularization technique involving batch-norm statistics for fine-tuning the color distribution of our visualizations (Yin et al., 2020). We study recurrent and feed-forward networks trained on ImageNet and CIFAR-10. More details on the networks, hyperparameters and regularization terms can be found in Appendix A.6.

Figures 7 and 8 show the results for ImageNet and CIFAR-10 datasets, respectively. Note that in these figures, the column of visualizations from recurrent models show the input-space image that activates output for the same filter but at different iterations of the recurrent module. These experiments show qualitatively that there are no inherent differences between the filters in the shallow layers and deep layers. In other words, the same filter is capable of detecting both simple and complex patterns when used in shallow and deep layers. For more visualizations and larger images, see Appendix A.7.

## 6 DISCUSSION

With the qualitative and quantitative results presented in our work, it is clear that the behaviour of deep networks is not the direct result of having distinct filters at each layer. Nor is it simply related to the number of parameters. If the specific job of each layer were encoded in the filters themselves, recurrent networks would not be able to mimic their feed-forward counterparts. Likewise, if the trends we see as we scale common network architectures were due to the expressiveness and overparameterization, recurrent networks should not experience these very same trends. This work calls into question the role of expressiveness in deep learning and whether such theories present an incomplete picture of the ability of neural networks to understand complex data.

In addition to comparing performance metrics, we show with visualizations that features extracted at early and late iterations by recurrent models are extremely similar to those extracted at shallow and deep layers by feed-forward networks. This observation confirms that the hierarchical features used to process images can be extracted, not only by unique specialized filters, but also by applying the same filters over and over again.

The limitations of our work come from our choice of models and datasets. First, we cannot necessarily generalize our findings to any dataset. Even given the variety of image data used in this project, the conclusions we make may not transfer to other image domains, like medical images, or even to other tasks, like detection. With the range of models we study, we are able to show that our findings about recurrence hold for some families of network architectures, however there are regimes where this may not be the case. We are limited in making claims about architectures that are very different in nature from the set that we study, and even possibly in addressing the same models with widths and depths other than those we use in our experiments.

Our findings together with the above limitations motive future exploration into recurrence in other settings. Analysis of the expressivity and the generalization of networks with weight sharing would deepen our grasp on when and how these techniques work. More empirical investigations covering new datasets and additional tasks would also help to broaden our understanding of recurrence. In conclusion, we show that recurrence can mimic depth in several ways within the bounds of our work, i.e. the range of settings in which we test, and ultimately confirm our hypothesis.

# 7 ETHICS AND REPRODUCIBILITY

We do not foresee any ethical issues with the scientific inquiries in this paper. In order to reproduce any of the experiments, see the code in the supplementary material and Appendix A.2. These materials are freely available and include all the details on training and testing, including specific hyperparamters, that one could need to run our experiments.

## ACKNOWLEDGEMENTS

This work was supported by the AFOSR MURI program, and ONR MURI program, DARPA GARD, and the National Science Foundation DMS program.

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

# A Appendix

## A.1 Model Architectures

We train and test MLPs, ConvNets, and residual networks on three image classification datasets. These models are described in Section 2, but the specific widths used for each dataset are detailed below.

**Multi Layer Perceptrons (MLPs).** All of the MLPs have a single fully connected layer before the internal module, and a single fully connected layer afterward. Each layer in an MLP can be defined by a matrix dimension. See Table 4.

Table 4: MLPs: The size of each fully connected layer in the MLPs we use for image classification.

| Dataset | First Layer | Internal Layers | Last Layer |
|---------|-------------|-----------------|------------|
| CIFAR-10 | $3072 \times 200$ | $200 \times 200$ | $200 \times 10$ |
| SVHN | $3072 \times 500$ | $500 \times 500$ | $500 \times 10$ |
| EMNIST | $3072 \times 500$ | $500 \times 500$ | $500 \times 47$ |

**ConvNets.** The ConvNets have two convolutional layers before the internal module and one convolutional and one fully connected layer afterward. Each convolutional layer is followed by a ReLU activation, and there are pooling layers before and after the last convolutional layer. For all convolutional layers outside of the internal module, we use $3 \times 3$ filters with stride of 1 and no padding. In order to preserve the exact shape of the inputs, the internal module has convolutional layers with $3 \times 3$ kernels, a stride of one pixel, and padding of one pixel in each direction.

Table 5: ConvNets: The number of channels in the output of each layer.

| Dataset | First Layer | Second Layer | Internal Layers | Last Conv Layer | Linear Layer |
|---------|-------------|--------------|-----------------|-----------------|--------------|
| CIFAR-10 | 32 | 64 | 64 | 128 | 10 |
| SVHN | 32 | 64 | 64 | 128 | 10 |
| EMNIST | 32 | 64 | 64 | 128 | 47 |

**Residual Networks.** The residual networks employ convolutional layers with $3 \times 3$ kernels. The first layer has a stride of two pixels and padding of one pixel in every direction. Each layer in the internal modules has striding by one pixel, while the final convolutional layer strides by two pixels. There are ReLU activations after every convolution, and average pooling before the linear layer. All the convolutional layers output 512 channels.

We also train and test residual networks with BatchNorm in order to compare the performance with ResNet-18 models. These residual networks have a unique batch norm layer after every convolutional layer followed by ReLU activation function. Since the input to the internal module at iteration $n$ is the output at iteration $n - 1$ (for $n = 2, 3, 4, ...$), the batch statistics for every iteration is different. Thus, we use separate batch norm layer for each iteration.

**Maze Solvers.** For solving mazes, we train and test residual networks with slightly different architectures. These models are fully convolutional and every convolutional layer has $3 \times 3$ kernels that stride by one pixel with padding of one pixel in each direction. They have one layer before the internal module, residual blocks of four layers in the internal module, and three convolutional layers afterward. All of the layers before the third to last one have 128 output channels, and in the final three layers the numbers of output channels are 32, 8, and 2.

## A.2 Hyperparameters

All the models are trained to convergence using suitable hyperparameters. For each model and each dataset, we fine tune the number of epochs, along with the learning rate and the decay schedule. For specifics, see the launch files in the code repository at <Suppressed for anonymity>.

### A.3   COMPUTE RESOURCES

All of our experiments are done on Nvidia GeForce RTX 2080Ti GPUs. All training, except for ImageNet models, is done on a single GPU. We train models on ImageNet data using four GPUs at a time. Each image classification model can be trained in several hours on a single GPU, and maze solving models require eight hours.

### A.4   ADDITIONAL VISUALIZATIONS

The discussion about activation patterns in Section 5 continues here with depictions of the activity in a ConvNet trained on SVHN and a residual network trained on CIFAR-10. See Figures 10 and 11. Although these plots show distinct behavior from Figure 6 in the body of this paper, the conclusions are consistent. In recurrent models, features are being reused. This is clear since every filter is being used on more than one iteration, and for many filters there is significant activity in most iterations.

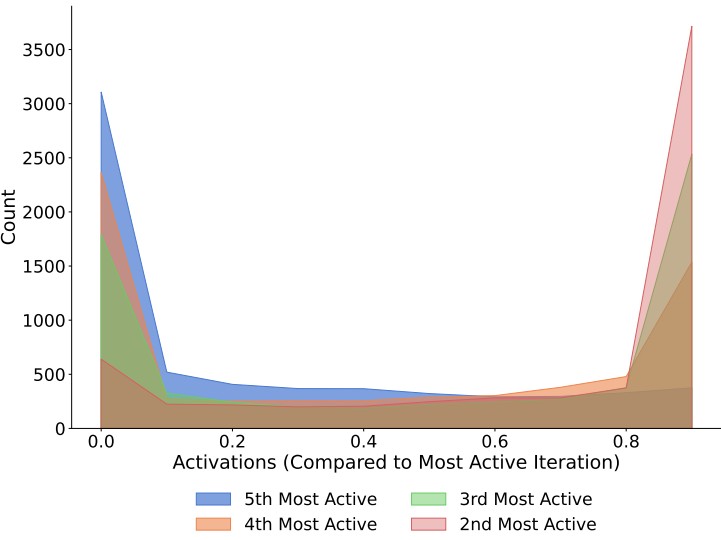

Figure 9: Relative activations at each layer in the recurrent module of an untrained CNN. This plot confirms that the distributions in the other similar plots from trained models are significant.

### A.5   CLASSIFIABILITY OF FEATURES

Compare the accuracy when a linear classifier is trained on features form different layers. As Kaya et al. (2019) do in their study of overthinking, we use average pooling to reduce the size of the features in this experiment.

Table 6: **Classifiability of Features.** CIFAR-10 classification accuracy of linear classifiers trained on intermediate features output by each residual block (or each iteration) in networks with 19-layer effective depths. Average pooling is used to change the dimension of feature maps in this experiment. This is consistent with (Kaya et al., 2019). Each cell reports the percentage of training/testing images correctly labeled.

|  | Block #1 | Block #2 | Block #3 | Block #4 |
|---|---|---|---|---|
| Feed Forward | 85.7 / 83.4 | 94.1 / 88.0 | 99.4 / 90.2 | 99.8 / 90.8 |
| Recurrent | 88.1 / 84.1 | 96.9 / 88.4 | 99.7 / 89.8 | 99.9 / 90.4 |

Additional experiments on the linear classifiability of features in recurrent models shows that other architectures are also separating features with each iteration.

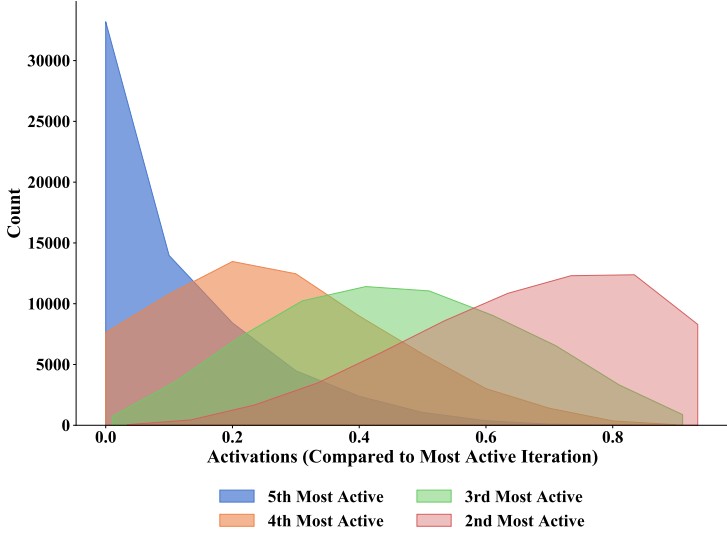

Figure 10: Relative activations at each layer in the recurrent module of a CNN trained on SVHN.

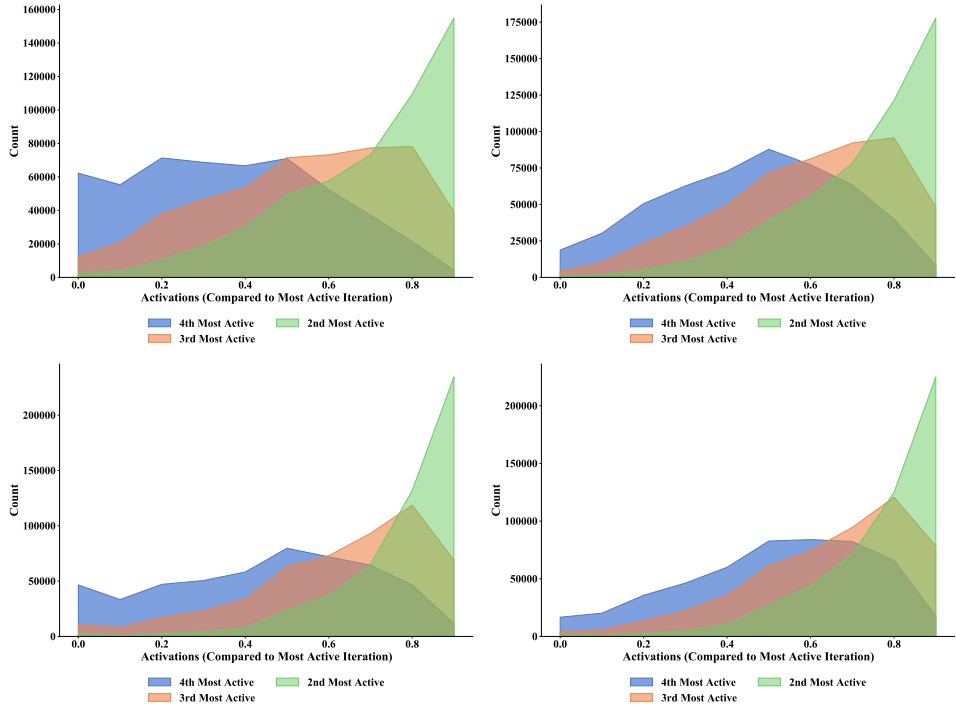

Figure 11: Relative activations at each layer in the recurrent module of a residual network trained on CIFAR-10.

### A.6 FEATURE VISUALIZATOIN

We discuss the feature visualization technique we used in 5.2 in further detail. For both ImageNet and CIFAR-10 datasets, we use networks with residual connections.

Similar to (Yin et al., 2020), we use pixel jitter as augmentation and total variation as regularizer. The pixel jitter moves the input image both horizontally and vertically by selecting two random number between $(-32, +32)$ for each axis. The portion of image that moves out of the viewpoint can fill up

Table 7: **Classifiability of Features.** CIFAR-10 classification accuracy of linear classifiers trained on intermediate features output by each iteration. Each cell reports the percentage of training/testing images correctly labeled.

|        | Layer #1 | Layer #2 | Layer #3 | Layer #4 | Layer #5 | Layer #6 |
|--------|----------|----------|----------|----------|----------|----------|
| R-MLP  | 46.52/44.28 | 50.67/47.22 | 57.54/52.07 | 59.43/54.08 | 61.43/55.19 | 61.54/55.39 |
| R-CNN  | 58.74/68.67 | 59.68/66.99 | 65.39/70.74 | 67.73/73.84 |          |          |

the empty space on the opposite side. For the total variation, we use the anisotropic version in four directions: horizontal, vertical, and two diagonal directions. We use the $\ell_2$-norm for computing the total variation in all directions.

We find that the batch statistics from the first convolutional layer provides a strong enough signal for improving the quality through regularization.

For CIFAR-10, we optimize a mini-batch of size $n = 64$ of $32 \times 32$ images for $1000$ iterations. For ImageNet we use a mini-batch of size $n = 32$ of $224 \times 224$ images for $1000$ iterations. Both experiments are done on a single Nvidia GeForce RTX 2080Ti GPU.

As the main loss for CIFAR-10 experiments, we maximize the $\ell_2$-norm of activations. More precisely, assuming that $a_{l,f}(x_i)$ represents the activation for filter $f$ in layer $l$ on input $x_i$, then we solve:

$$\max_x \sum_i \|a_{l,i}(x_i)\|_2$$

For ImageNet experiments, use a contrastive version of the loss to promote diversity:

$$\max_x (\sum_i n\|a_{l_i}(x_i)\|_2 - \sum_{j \neq i} \|a_{l_i}(x_j)\|_2)$$

We use the ADAM optimizer and cosine annealing based learning rate schedule. The initial learning rate is $0.01$.

Assuming that $\lambda_{main}, \lambda_{tv}, \lambda_{bn}$ represent the coefficients for the main loss, total variation and batch normalization regularizer, respectively, we set $\lambda_{tv} = 0$ and $\lambda_{bn} = 0$ for the visualizations in the first layer, and linearly increase them as we go deeper. For the last layer, in CIFAR-10 experiments, $\lambda_{main} = 0.64$, $\lambda_{tv} = 1.0$, and $\lambda_{bn} = 10$. For ImageNet experiments, in the deepest layer we set $\lambda_{main} = 0.01$, $\lambda_{tv} = 5.0$, $\lambda_{bn} = 50.0$.

## A.7 Extensive Qualitative Visualization

Similar to Section 5.2, here we show the results of our visualizations in Figures 12 and 13. We note that the first and final layers are not part of the recurrent modules.

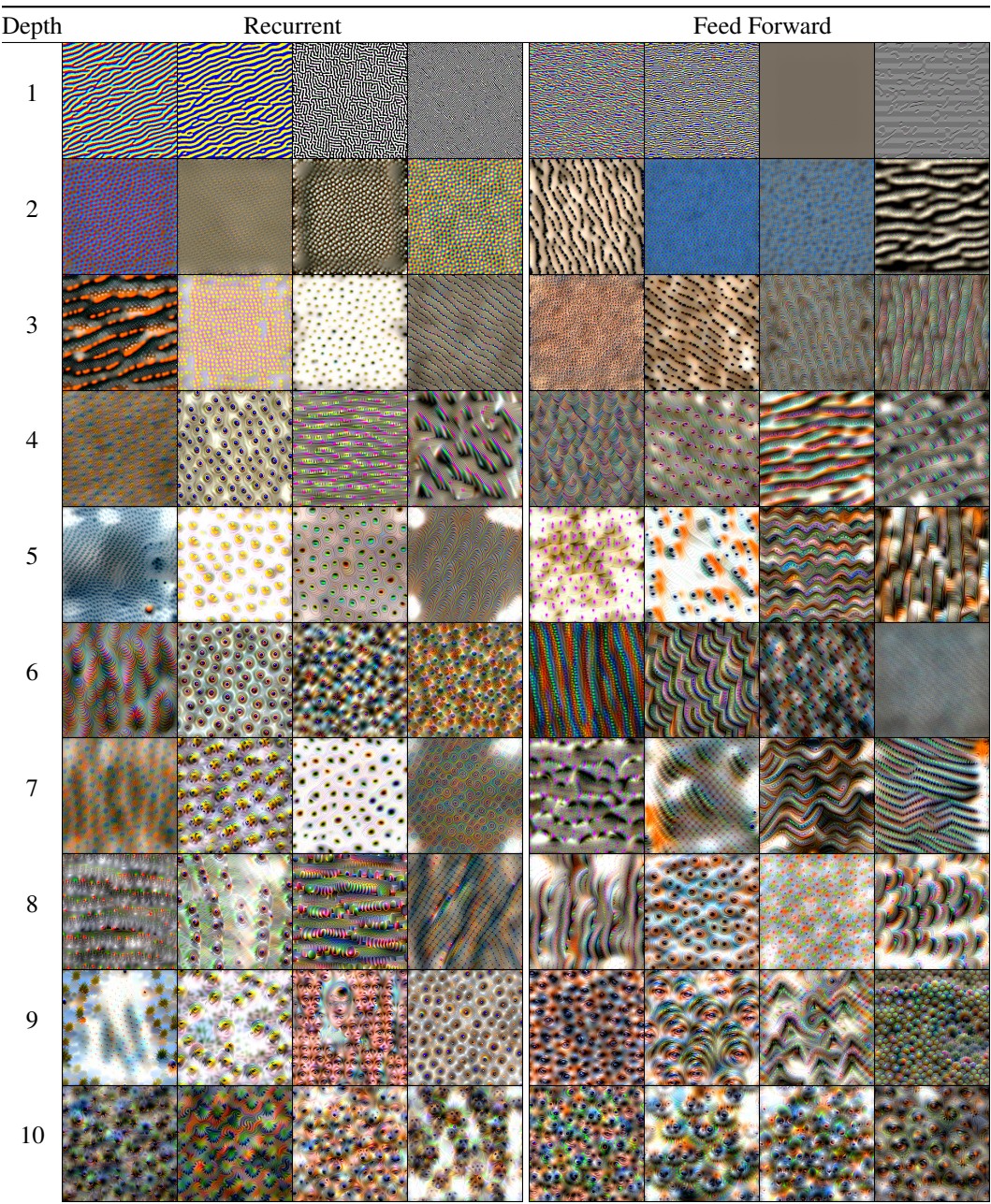

Figure 12: ImageNet filter visualization. Filters from recurrent model iterations alongside corresponding feed-forward layers.

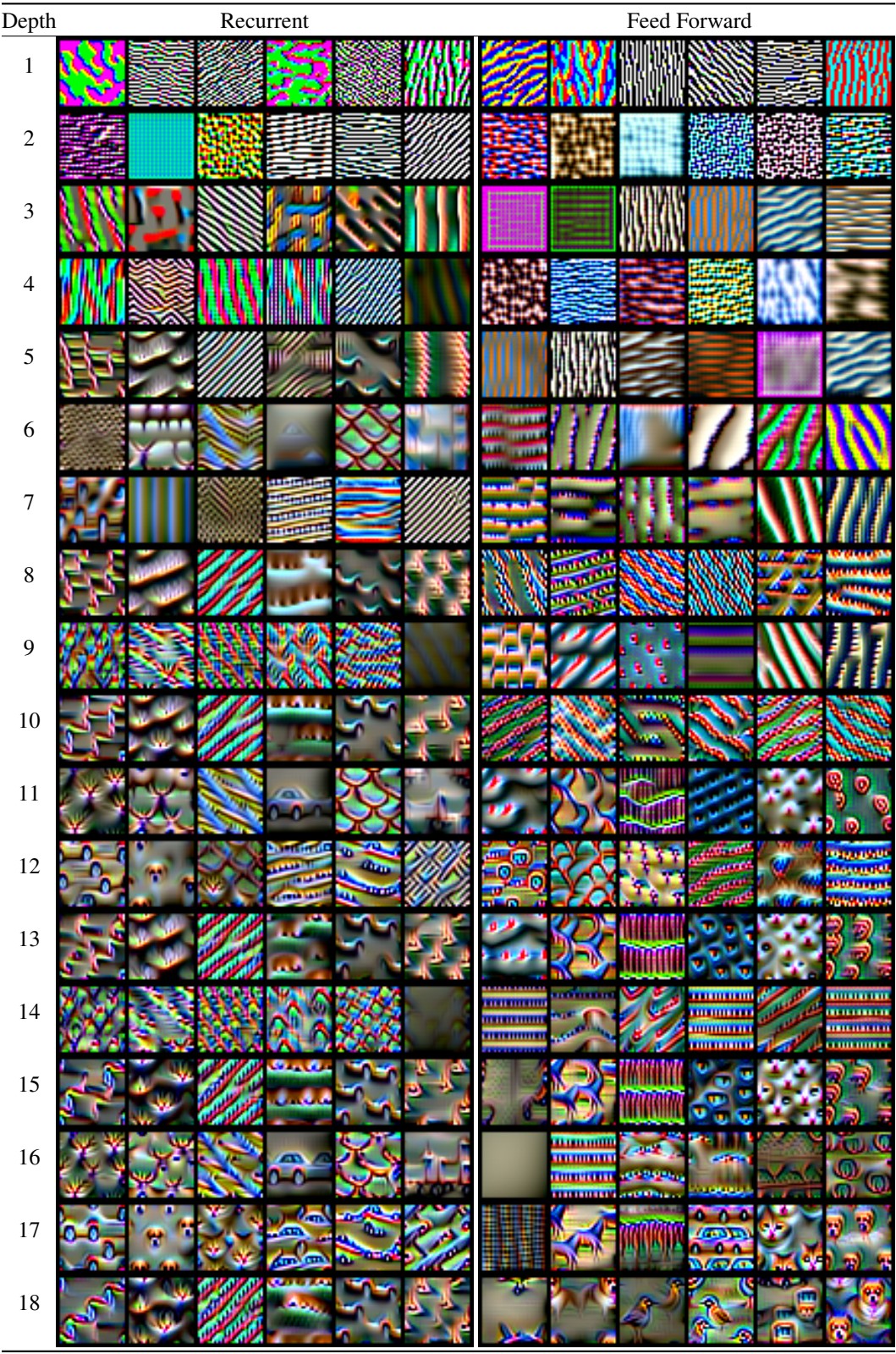

Figure 13: CIFAR-10 filter visualization. Filters from recurrent model iterations alongside corresponding feed-forward layers.

