# OpenReview forum: "The Uncanny Similarity of Recurrence and Depth"
_ICLR.cc/2022/Conference — ICLR 2022 Poster_

### Official Review · Reviewer_opFT · 2021-10-31

**Correctness:** 3
**Technical Novelty And Significance:** 3
**Empirical Novelty And Significance:** 3
**Recommendation:** 8
**Confidence:** 4

**Main Review:**

# Strong points

-   The paper is clear and easy to understand
-   The authors conclude multiple experiments to support their claims
-   They use multiple datasets, including ImageNet
-   They use multiple network families, including MLPs, CNNs and ResNets
-   They also use linear probing of layer-wise representation power, which support their claims
-   Feature visualisation also supports their finding
-   The correspondence of feature visualisations, linear predictive power and general trends is very surprising.

# Weaknesses

-   The minimal effect of weight sharing was already shown in some cases [1,2], although this paper analyses it much more thoroughly and shows a broader picture of the effect
-   There are some indications that the similarity would break down with deeper networks.
-   The analysis of feature reuse can be because of different effects (Section 5, first part).
-   For filter visualisations, it is unclear whether the same neurons are responsible for high and low-level features

Please find more details below.

# Questions/suggestions
-   For the maze problem, it is unclear how the performance is measured. Is it pixel-wise, or the whole maze should be correct for the sample to be considered correct?
-   For the ResNet architectures, the parameters of the batch norm are not shared between the layer. But multiple papers are suggesting that these parameters can have a significant effect on the behaviour of the network [3,4,5]. Therefore, it would be worth considering making a partially shared variant of the batch norm that shares the affine parameters but not the statistics.
-   The analysis of the activation strengths (Section 5, first part) might not indicate what one might think. If there are biases in the network, the outputs of specific neurons can be high even when “they are not doing anything”. Also, it is questionable whether having high activation most of the time means that they are actually doing computation most of the time. The actual information might be carried on top of a high bias. That said, I don’t have a good suggestion on how this can be adequately measured. Maybe measuring variance between the different samples might be a better metric. Probably visualisation of the same unit in different depths would be informative. Maybe soften the claim in a bit when discussing.
-   For feature visualisations, it is unclear whether the low-level and high-level features shown in the figures come from the same or different neurons for feature visualisations. How does the presence of the high and low-level features change through the layers? Figure 12 shows high-level features together with low-level ones already at depth 11. It would be nice to visualise the same neuron over “depth”.
-   It would be nice to cite [6] when discussing feature visualisations in Section 5.2. It is an outstanding analysis of how networks work.

# Additional suggestions that could make the paper better
-   In Figure 2, the gap starts growing noticeably with deeper networks (19 layers on CIFAR-10, EMNIST, ImageNet). It seems like the RNNs overfit sooner than the CNNs (CIFAR-10, EMNIST), which is the opposite that I would expect. It would be worth investigating this effect more.
-   The ImageNet experiment uses a very small ResNet and with 19 layers. Together with the previous point, I would be curious about the gap with a more realistically-sized ResNet.

[1] Liao et al: Bridging the Gaps Between Residual Learning, Recurrent Neural Networks and Visual Cortex (2016)
[2] Alexandre Boulch: ShaResNet: reducing residual network parameter number by sharing weights (2017)
[3] Revisiting Batch Normalization For Practical Domain Adaptation (2017)
[4] Frankle at al.: Training BatchNorm and Only BatchNorm: On the Expressive Power of Random Features in CNNs (2020)
[5] Kanavati et al.: Partial transfusion: on the expressive influence of trainable batch norm parameters for transfer learning (2021)
[6] Cammarata et al.: Thread: Circuits (2020)

**Summary Of The Paper:**

The paper shows that sharing the weights between inner layers of image classification networks has minimal effect on their performance: they achieve similar accuracy with significantly fewer parameters. This questions the commonly accepted view that later layers of CNNs function as more and more specialised filters. It also suggests that instead of the number of weights, the amount of computation might be the main reason behind the good performance of the deep models. Interestingly the authors use feature visualisation to show that although the weights are shared, the network still learns hierarchical features.

**Summary Of The Review:**

I recommend accepting the paper because it highlights the importance of computation depth instead of the number of parameters (which is commonly assumed to be a good indicator of the predictive power). This might motivate further research in models with increased computation depth. Despite the weaknesses, the authors presented enough evidence for supporting the main messages of the paper.

---

> ### Author Response · Authors · 2021-11-22
> **Response to Reviewer opFT**
>
> Thank you for the positive feedback! For Mazes, the entire output must match in order to be counted as correct. We have updated the draft to clarify. In none of our networks is the bias turned on in the convolutional layers. To strengthen the feature reuse section, see the additional plot of the activations for an untrained network in the appendix and the newly added discussion of feature space similarity metrics. Lastly, our ResNet-50 gets 76.48% correct on ImageNet, and a recurrent version with weight-sharing within stages achieves accuracy of 75.13%. See the comment to reviewer 9nUr for more information.

---

### Official Review · Reviewer_9nUr · 2021-11-03

**Correctness:** 3
**Technical Novelty And Significance:** 1
**Empirical Novelty And Significance:** 2
**Recommendation:** 6
**Confidence:** 5

**Main Review:**

Strengths
- **Intriguing Topic**. Deep learning papers, lectures, and even books at this point all underline the importance of parameters in the success of deep networks as compositions of learned and nonlinear functions. Scrutinizing this more closely, and questioning if its the sheer number of parameters that matters, is of definite interest. Existing work provides evidence that recurrence across depth/shared weights can suffice, but a paper that thoroughly studies shared/unshared layers could draw more attention to the topic. This is true whether the results reinforce or erode the existing intuitions!
- **Variety of Measures**. While task accuracy is studied first, and should be, further experiments examine processing through other measurements. For example, depth is contrasted with receptive field size (for mazes, Sec. 4.2), activation magnitudes are compared across layers (Figure 6), and discriminability is checked layer-by-layer (Table 3). Visualization is also used to qualitatively compare filter responses across recurrent and feedforward models (Figures 7 & 8).
- **Clear Writing**. The introduction is direct and concise in justifying the research question of whether and how models with shared weights differ from models with distinct weights. The setup (Sec. 2) defines its terms and goals, with sufficient detail on the architectures studied to reproduce the experiments. The figures and tables are legible and appropriately captioned.

Weaknesses
- **Missing Related Work**. The related work is incomplete and does not sufficiently credit existing demonstrations of networks that are recurrent in depth. [A] analyses the effects of sharing weights across depth for convolution and contrasts it with other design choices. [B] defines and experiments on partially recurrent residual networks, and addresses the same point about normalization as Section 3.1 in this work. [C] explores aggressive sharing of weights across depths in residual networks and compares with standard architectures at common scales of model depths and parameters. More afield, but not unrelated, are methods that unroll an optimization process and show that sharing weights can be effective, such as [D]. This work could provide more in-depth analysis, or more up-to-date analysis w.r.t. the models considered, but prior work must be credited, above all in case the earlier work can inspire more analysis.
- **Narrow Experimental Scope**. The experiments are restricted to relatively shallow nets (with depth of 19 or less) in the visual domain. The chosen architectural constraints, like no intermediate pooling or change in channel dimensions, do conflict with standard networks. Experiments that control for these are valuable, and are included in this work, but the experiments should also examine standard architectures. At the current scope, I am concerned this work runs the risk of confusing matters, while a broader look would be more clarifying for the community. To start, experiments could cover deeper nets, and then go on to look at other tasks like sequence modeling (as done by implicit nets, which also share weights).
- **Technical Redundancy**. Recurrent nets of this type have already been defined and experimented on, as done by [A, C] and partially [B]. The prior existence of such networks leaves this work without technical novelty. To be clear this is not a fatal flaw, but it does squarely place the value of this work in the breadth and depth of its empirical results.
- **Task Choice**. The maze task is non-standard, and therefore less informative than any other established task, although it does make some sense. The analysis does support its soundness by showing that there is a distribution of difficulties, that depth helps across them, and that receptive field size does not fully explain away the results. However, all of this could be spared by adopting a task like semantic segmentation on PASCAL VOC or Cityscapes, which are both well-known, and even image-to-image tasks not unlike the maze task.

For Rebuttal
- **Related Work**. Please position this work relative to the papers raised in this review, especially [A, B]. Please also situate the contributions of this work w.r.t. the implicit models in work on DEQs and NODEs that show such models can rival explicit models, and even include purely recurrent models (as ablations of their inference optimization) as baselines as in Bai et al. 2020.
- **More Depth**. Please experiment at depth=50 for the results reported in Figure 2. Please likewise experiment with a standard ResNet-50 architecture and compare it to a recurrent analogue, such as a net that shares the parameters across blocks in each stage (but not the normalization layers, as higlighted by Section 3.1).
- **Reuse Analysis**. Please discuss other options for measuring reuse, and furthermore interventions to check the analysis. For instance, what if channels are masked out (that is, multiplied by zero) at their least active layer? Severe loss in accuracy from this masking would provide stronger evidence that the reuse is happening and necessary. In the same vein, what does Figure 6 look like for an untrained network? If it differs, this would support the use of the proposed analysis by counting.

Miscellaneous Feedback
- [clarity] Please define "depth" up front, in the introduction. The definition at the beginning of Section 2 is good, but a one sentence summary in the first paragraph or at least on the first page would orient the reader more immediately.
- [format] Please consider sharing the y axes across the plots in Figure 2 to consistently convey the size of the differences.
- [method] For normalization (Table 1), consider sharing the parameters—but not the statistics—of the batch norm layers. Sharing the affine parameters would make the model even more recurrent, while still allowing for different statistics across recurrent iterations. In the same vein, consider a comparison with group norm, which is known to work with ResNets, but unlike batch norm has purely local state.
- [clarity] The neuroscience reference in the discussion comes out of nowhere, and would be more illuminating if its content was concretely connected to the architectural choices and experimental results in this paper.
- [related work] As a possible counterpoint to theory of layer specialization, you may be interested in reading https://arxiv.org/abs/1605.06431 for its experiments that question the importance of any one layer in a residual network.

References
- A. Understanding Deep Architectures using a Recursive Convolutional Network. Eigen et al. arxiv'13 and ICLRW'14.
- B. Residual Connections Encourage Iterative Inference. Jastrzębski et al. ICLR'18
- C. ShaResNet: reducing residual network parameter number by sharing weights. Boulch. arxiv'17 and PRL'18
- D. Conditional random fields as recurrent neural networks. Zheng et al. CVPR'15

**Summary Of The Paper:**

This work examines what constitutes depth for deep networks by examining inference for networks with and without distinct parameters at each layer. Sharing weights across layers defines networks that are recurrent in depth, and still compose the same number of nonlinearities, but reduce the number of parameters. Recurrent architectures of this kind challenge a common intuition about hierarchy in deep representations and so deserve analysis. The experiments in this work carry out controlled comparisons between pairs of networks with shared/unshared parameters, and measure: (1) task accuracy, (2) depth vs. receptive field, (3) activation statistics across layers, (4) discriminability of features across layers, and (5) visualization of filter responses. The tasks studied are image classification, with standard datasets like ImageNet and CIFAR-10, and a custom task concerning images of 2D mazes. The architectures control for differences between the feedforward/unshared and recurrent/shared models, but exclude standard architectures as references. The networks in these experiments are relatively less deep (at 19 or fewer layers), and differ from standard networks (like a ResNet-50) in their lack of pooling and fixed channel dimensions. As suggested by the title, the results for a given depth are indeed quite similar with or without recurrence, at least in most cases for the networks and tasks studied. In this scope, the work convincingly shows that distinct parameters are not necessary, but that does not necessarily generalize to more standard deep networks for vision, nor to other modalities of data, nor to other prediction tasks.

**Summary Of The Review:**

It is nice to see this work take a closer look at what exactly differs or not across layers with distinct or shared parameters. This is an important topic to scrutinize since deep models are now so common while sharing weights across depth is still rather rare. The intuitive or "folkloric" understanding of deep networks as strictly hierarchical is not beyond empirical analysis, and so in general I welcome this effort. Nevertheless there is more to be done for this work to be sufficiently thorough and informative. Putting aside the missing related work, and considering only the content in the paper, the analysis done here is quite narrow. The only modality considered is vision, the only mainstream task is classification, and standard/unshared models are only studied where they most closely align to recurrent models. Inference is studied in several aspects, like accuracy, feature visualization, and layer-wise prediction, but optimization is not addressed at all. That is, does a recurrent model train differently than its paired feedforward model, even if they ultimately converge to similar accuracy? In summary, I encourage the authors to continue in this direction but with a wider lens, and re-submit with more coverage of different data and tasks, and most of all broaden the bounds of which shared vs. unshared architectures they consider (for instance, deeper nets like ResNet-50 due to its popularity, and perhaps hybrid architectures that still have pooling but share all weights within a "stage" at a given resolution, and so on.).

**Final Review** The author response addressed the points requested for rebuttal: inclusion of missing related work, an experiment at greater depth with greater resemblance to a standard ImageNet classifier, and more justification of the layer-wise activation analysis. While I still find the technical and empirical novelty of this work to be lacking, as much or all of it can be reconstructed from the related work, I can see value in the unified experimental setup and the juxtaposition of different results. As this paper could bring more attention to recurrence and depth, as earlier work seems not to have done, it could inform the community. I have therefore raised my score to 6.

---

> ### Author Response · Authors · 2021-11-22
> **Response to Reviewer 9nUr**
>
> On the issue of more depth, we have now trained a standard ResNet-50 and a weight-shared (within stages) version on ImageNet. Our ResNet-50 gets 76.48% correct on ImageNet, and weight sharing within stages yields accuracy of 75.13%, verifying that even large standard ImageNet models exhibit performance similar to their recurrent counterparts which possess far fewer free parameters. Training models without striding and pooling (like our fully weight-shared architectures) is not feasible at this depth. We have added this result to the section on ImageNet.
>
> A figure of the activation comparisons of an untrained net is now available in the appendix. We have also now measured feature space similarity metrics.
>
> Thank you for listing missing relevant work, we have now better contextualized our work in the updated related work section.
>
> *Miscellaneous Feedback* Thanks for these pointed suggestions. We have now clarified the use of depth in this paper in the first paragraph. Also, after careful consideration, we have added a note in the caption drawing attention to the fact that the scales are different, but we think the plots better convey the information when the axes are chosen per dataset as the accuracy scales differ significantly between datasets so that curves on some datasets (such as EMNIST or SVHN) would look flat if the y-axis scale is big enough to accommodate harder datasets such as ImageNet. We have now revised the discussion section.

---

> > ### Comment · Reviewer_9nUr · 2021-11-30
> > **Thank you for the response w.r.t. related work, depth, activation analysis, and the feedback on the exposition.**
> >
> > The response has addressed the points requested for rebuttal: inclusion of missing related work, an experiment at greater depth with greater resemblance to a standard ImageNet classifier, and more justification of the layer-wise activation analysis. While I still find the technical and empirical novelty of this work to be lacking, as much or all of it can be reconstructed from the related work, I can see value in the unified experimental setup and the juxtaposition of different results. As this paper could bring more attention to recurrence and depth, as earlier work seems not to have done, it could inform the community. I have therefore raised my score to 6.
> >
> > Please consider citing Boulch regarding the recurrent ResNet-50-like model reported in this response, if for nothing else than inspiration, as comparable models are experimented with in that work. Of course, as pointed out, the focus there is on reducing parameters but it is nevertheless relevant.
> >
> > In particular I want to acknowledge the improved revision of the related work and discussion, which now better situate the work and its findings with respect to what has already been done (and not). I believe this will make the work better received, and hopefully have more impact in drawing more interest to the analysis of shared vs. unshared weights in deep models.

---

### Official Review · Reviewer_UD8X · 2021-11-03

**Correctness:** 3
**Technical Novelty And Significance:** 2
**Empirical Novelty And Significance:** 2
**Recommendation:** 6
**Confidence:** 4

**Main Review:**

#############

UPDATED REVIEW:

Thank you answering all my questions and apologies, for the delayed response.

Upon reading the rebuttal and revised draft, I believe the authors have satisfactorily addressed my concerns and hence, I am increasing my score by 1 point. I would be interested, however, to see how the hierarchical representation plays out in the future with different architectures and more complex tasks!

#############

Strengths: The paper asks an interesting question on whether depth in artificial neural networks can be replaced with recurrence that would use much fewer parameters. The analyses on qualitatively visualizing the learned filters was exciting and insightful in how these models represent information/learn hierarchical structure even with recurrence.

Weaknesses: My comments and questions are listed below.

Image classification task:
I am not sure I find this result particularly surprising. In both MLP and R-MLP, there are additional non-linear transformation on the model inputs and this leads to some patterns in accuracy. Granted that the “inputs” to the middle layer are recurrent in one case and not the other, which has implications on the number of model parameters. But in several cases:

1. Are the differences between the effective depth across a model architecture statistically signifiant to begin with? If not, I am not sure it is fair to claim that the recurrent versions mimic the non-monotonic patterns of depth. Instead, they just achieve similar accuracy to the original model

The above comment is in reference to:
> First, the trend in performance as a function of effective depth is not always monotonic and yet the effects of adding iterations to recurrent models closely match the trends when adding unique layers to feed-forward networks.

> In Figure 2, the striking result is how closely the blue and orange markers follow each other.

For example, in the MLP based models, it is not clear if the differences across depth are significantly different from each other.

2. The biggest differences in accuracy seem to arise from different architectures themselves and not the absence or presence of recurrence. This further suggests that the types of functions approximated by the model (either as stacked non-linearities or recurrent ones) is more important.

3. For ImageNet, we observe a monotonically increasing accuracy curve as the number of non-linearities in the model increase (irrespective of how they increased). What conclusions do the authors draw from this aside from the following?
> These experiments show that the effect of added depth in these settings is consistent whether this depth is achieved by iterating a recurrent module or adding distinct layers.

While the results are demonstrative that using recurrence (and hence, fewer parameters) one can achieve similar task performance on image classification, without a theoretical proof, I do not think they indicate that the models learn similar functions (I understand the authors never claim this but perhaps I'm missing the bigger/less obv picture painted by the generalization performance result).

Maze task: The dataset itself is very interesting! As a minor point, the authors mention that fewer than 0.5% of the mazes are duplicated within each of the training sets. What is this statistic for the train-test overlap? I also found the linear probing analyses in section 5.1 to be demonstrative of the increasingly separable features learned as (pseudo-) depth increases.

My comments and questions in the image classification task apply here as well.

To me the most exciting part of the paper was the analysis in section 5.2. While it seems obvious to me that accuracy improves as depth increases (whether or not the parameters are reused), the finding that qualitatively these models learn similar features despite using different mechanisms of weight sharing is insightful. It suggests that the information captured by neural networks is possibly not in the parameters themselves but the states of the model. However, I would be interested in learning the authors' thoughts on whether this is expected given the increasingly non-linear nature of the input to the model. Further, can the authors comment on the lack of alignment in the recurrent and feed-forward columns visualized in Fig. 12?

The paper would greatly benefit from adding theoretical proofs to suggest that these models are perhaps approximating similar functions or adding a wider suite of complex tasks & architectures. See for example this paper that theoretically and empirically proves recurrence and depth are equivalent in RNNs: https://arxiv.org/pdf/1909.00021.pdf

In its current form, I am not convinced the generalization results are surprising/insightful given that both version of the models have access to stacked non-linearities but I encourage the authors to explore and emphasize on the hierarchical structure emulated by recurrence in the future.
> we find that recurrent networks can emulate both the generalization performance and the hierarchical structure of deep feed-forward networks

Other questions:
- What are the results for classifiability of features with other architectures? (table 3)
- How was hyper-parameter tuning done? What are the train-val-test splits for different datasets? How much time did each model take to train and test? Was there signifiant differences in optimizing one over the other?

**Summary Of The Paper:**

In this work, the authors compare and contrast models wherein layer depth is replaced with equivalent number of recurrent time steps. In particular, they explore feed-forward, CNN and residual deep network wherein the intermediate layers are replaced by an equivalent recurrent block. For each architecture, the authors compare accuracy of the stacked and recurrent variants for image classification on 4 datasets- CIFAR-10, EMNIST, SVHN and ImageNet and a novel Maze task wherein the network has to assign binary labels to each image pixel (0= not a valid point in path, 1=valid point in path to solve the maze). Overall, the authors observe that across all models and tasks, the recurrent and stacked networks achieve similar performance. They also verify this with batch normalization and dilated CNNs in some tasks. Next, they investigate the number of positive activations at each recurrent step to find that filters are active/recycled across recurrent depth. They also analyze the linear classification accuracy at different depths of the stacked and recurrent CIFAR-10 feedforward model, finding comparable accuracy at each depth. Finally, they visualize filter activations at different depths for both CIFAR-10 & ImageNet, finding comparable feature maps in both architectures.

**Summary Of The Review:**

The paper has interesting results on how hierarchical representations emerge in recurrent networks akin to stacked networks, despite the former sharing parameters across its "depth". However, the similarity in task accuracy between recurrent and stacked networks with similar number of non-linearities is not terribly surprising. It also doesn't shed light on whether these models approximate similar functions on whether this finding can be generalized to other task paradigms. I believe this is a promising research direction and encourage the authors to analyze the similarities and differences in the information learned by the two paradigms in more depth.

---

> ### Author Response · Authors · 2021-11-22
> **Response to Reviewer UD8X**
>
> Thank you for the thoughtful review.
>
> In Figure 2, horizontal bars indicate one standard error above and below the mean of our trials. These tight intervals indicate that the performance differences are significant.
>
> We agree that if the goal is to achieve higher accuracy on the tasks we consider, the model architecture is a huge factor. Rather than improving performance, our aim is to show that in many cases, there is little qualitative or quantitative difference between adding recurrences and increasing depth. With this in mind, it is consistent with our thesis statement that bigger differences arise with changes in architecture rather than with the absence/presence of recurrence.
>
> Within the range of depths we study, depth helps on ImageNet whether the parameters are shared or not. Our choice of depths is based on what is feasible to train for RNNs on ImageNet, and the monotonic trend observed is in line with the fact that SOTA models on this dataset tend to be very deep.
>
> The maze training and test datasets do not overlap at all.
>
> Figure 12 shows several examples where the feature visualizations are qualitatively similar. While finding two neurons that lead to exactly the same visualization is very unlikely, this figure shows that the types of inputs that excite early/late layers are visually similar in recurrent and feed-forward neural networks.
>
> The recurrent CNNs and MLPs also separate features as they iterate. See Table 7 in the appendix.
>
> The training and testing splits of the vision datasets are according to their conventional splits (CIFAR, MNIST, EMNIST are available through Pytorch, and ImageNet is the ILSVRC2012 dataset, and we test on the official validation split). Tuning hyperparameters is done with a coarse grid search over the learning rate, decay factor, decay schedule, and number of epochs. Tuning took approximately the same number of runs for each model/dataset, and training times vary with the numbers of epochs. The exact hyperparameter values for our experiments are available in the code, which is in the supplementary material and will be made publicly available online after the review process.

---

### Official Review · Reviewer_Ej1w · 2021-11-08

**Correctness:** 3
**Technical Novelty And Significance:** 2
**Empirical Novelty And Significance:** 3
**Recommendation:** 6
**Confidence:** 5

**Main Review:**

Pros:
- I believe that the authors are studying an interesting, although not new, question about the similarity between recurrence and depth in information processing in neural networks. There has been immense progress in the field of AI (particularly computer vision) with feedforward models and it is puzzling why recurrent models haven't caught up (despite being computationally sophisticated and present in abundance in the cortex). This work is sparking further interest in studying the relationship between depth and recurrence.
- The authors conduct extensive quantitative and qualitative analyses with several model families and diverse effective depth ranges to test their hypothesis that recurrence mimics depth.
- The newly proposed maze challenge is very interesting and connects well to prior work in Vision science and psychophysics that study the influence of global cues in visual processing.
- I find Section 5 on recurrent models reusing features to be interesting, but the clarity of writing in this section could be improved for better understanding on the readers' end.

Cons:
I have the following concerns about this work:
1. How are the findings in this work different from Liao and Poggio 2016 [1]? For e.g. Figure 3 in their paper conducts the same analysis that is conducted by Section 3 (albeit with a wider range of datasets). While the observed results confirm that deep architectures and recurrent architectures achieve similar classification accuracy, I didn't find this section particularly novel.
2. The studied recurrent models are very simplistic in nature (stack of convolutions with weight sharing) and do not incorporate several crucial components that make recurrence interesting (both in ANNs and in the brain) such as gating, bypassing, temporal decay, etc.
3. There isn't enough details about the recurrent architectures leading to following unresolved questions: Exactly how many layers are recurrent in each of these architectures? How many steps of recurrence is implemented in these layers?
4. The maze solving task is interesting, however, could one explain the results merely based on the effective receptive field size growth with recurrence and depth? The authors say that RF size cannot explain the full effect as the accuracy increases beyond the point where models achieve enough RF size to capture the full maze, however this is confounded with increase in # parameters and capacity as the effective depth is increased. The authors themselves report results from a dilated convolution experiment showing superior performance with larger receptive fields at shallow network depths. Hence, I strongly feel that the similarity observed can be largely reduced to both recurrent and feedforward networks increasing their effective receptive field size with more depth / iterations.
5. I feel that the paper makes quite a few very strong claims in the Discussion section that aren't fully tested by the Methods. I feel that this section needs to be toned down further to promote more exploration into these areas through future experiments.

References
[1] Liao, Q., & Poggio, T. (2016). Bridging the gaps between residual learning, recurrent neural networks and visual cortex. arXiv preprint arXiv:1604.03640.

**Summary Of The Paper:**

The authors explore the similarity between neural networks that accrue information processing via depth and via recurrent iterations. They have conducted several interesting qualitative and quantitative analyses to study the above question and conclude that recurrent processing mimics depth in the experiments they have conducted and draw links to relevant work in computational and experimental neuroscience that also study the role of recurrence in information processing in the cortex.

**Summary Of The Review:**

I find the question approached by the authors to be an interesting (although not novel) one and requires studying from various angles in order to shed more light on how recurrence and depth are related. The authors have made a nice effort to perform various analyses (both quantitative and qualitative) to study their hypothesis that recurrence mimics depth of processing. However, I find part of this study (Section 3) to be quite similar to prior work in this area although scaled up with more datasets. In Section 4, I find contradictory claims such as the authors claiming that performance rise not fully explained by receptive field size, and contradicting this statement with models containing dilated convolutions (exponential receptive field growth with depth) that produce large performance gains. The recurrent models studied are quite simplistic and lack several interesting components that have contributed to the expressivity of recurrent processing in the past (such as gating, bypassing, nonlinearities, temporal decay etc.).

At the current stage, I feel that the readers do not gain much new information about the relationship between depth and recurrence beyond what has been established by prior work. I suggest the authors to experiment with more sophisticated recurrent networks and explore more novel analyses (such as the maze challenge, which I find to be really interesting but needs to be scaled up to explain further similarities beyond receptive field size growth) to understand the relationship between depth and recurrence. This is what I feel is required to truly expand our understanding of (whether, and under what conditions) recurrence mimics depth, and under what conditions / tasks the former is more expressive than the latter and vice versa.

======= Update after author response =======

I have read the authors' response and updated manuscript, after which I am increasing my score to 6. I believe that the updated manuscript's contributions are clear and builds on the interesting work from Liao and Poggio, 2016, adding further information with novel analyses and a new dataset (Maze challenge). I believe this work will spark further discussion and exploration of the similarity between recurrence and depth.

---

> ### Author Response · Authors · 2021-11-22
> **Response to Ej1w**
>
> Thank you for the pointed and constructive comments.
>
> We have now better contextualized our work by adding to our related works: “Similarly, Liao & Poggio (2016) show that deep RNNs can be rephrased as ResNets with weight sharing and even include a study on batch normalization in recurrent layers. Our work builds on theirs to further elucidate the similarity of depth and recurrence, specifically, we carry out quantitative and qualitative comparisons between the deep features, as well as performance, of recurrent and analogous feed-forward architectures as their depth scales. We analyze a variety of additional models, including multi-layer perceptrons and convolutional architectures which do not include residual connections.”  We also want to point out that we have now computed feature space similarity metrics, a quantitative analysis not performed in [1], and incorporated them into our draft.
>
> We have now revised Sections 2.1, 2.2, and 2.3, where we delineate the architecture in the recurrent portion of each architecture, to make the details more clear. MLPs and CNNs have single-layer recurrent modules that can be repeated; and the number of repetitions is indicated by the corresponding effective depth (which includes the non-recurrent layers). Residual networks have a whole residual block with four layers as the recurrent module.
>
> Thank you for the feedback on the discussion section. We have revised it to better summarize our findings, our limitations, and possible directions for future work.
>
> Gates etc. are interesting, however in our work we aim to isolate the impact of recurrence by comparing otherwise similar models. The inclusion of gates might introduce a confounding variable into our study, but is an interesting direction for future work.
>
> In most of our models, the receptive field is complete, i.e. covers the entire input, early in the network. At several depths we considered, there are many layers of processing after the RF covers the entire input. To disentangle the parameter count/capacity from the RF, we report results in Table 2. While the accuracies are higher -- which we agree shows that RF is important for maze solving -- the similarity of recurrence to depth persists even once the RF is complete. This makes a compelling case that *even* when RF is important, there is still performance gain with depth that RF can’t account for.

---

> > ### Comment · Reviewer_Ej1w · 2021-11-29
> > **Reply to author response**
> >
> > I thank the authors for their response to my comment. I believe that the additional clarifications and additions are really helpful to understand the contributions of this paper and relate it to prior art. The discussion section reads much more clear now and specifies contributions, room for improvement and limitations of the current work. I am hence updating my score to 6 after reading the authors' response and updated manuscript. I feel that the contributions of this paper builds on the interesting work from Liao and Poggio, 2016, and adds further information with novel analyses and a new dataset (Maze challenge) to spark further discussion and exploration of the similarity between recurrence and depth.

---

### Author Response · Authors · 2021-11-30
**General Author Response to Reviews**

We thank the reviewers for their time and their thoughtful feedback. We appreciate that the reviewers all recognize that our questions about recurrence and depth are interesting and valuable and that our empirical results on feature reuse in recurrent models are insightful.

Prompted by their feedback, we have updated our related works section, which now includes references to all the sources mentioned by reviewers, as well as a more complete picture of how our work fits with the existing body. We have also updated the draft to include architectural details as Reviewer Ej1w suggested. Finally, we have new results with deeper models on ImageNet, which were suggested by Reviewers 9nUr and opFT, as well as further experiments on linear separability with different architectures, as suggested by Reviewer UD8X.

We emphasize that in our effort to incorporate all suggestions, we have updated the draft and responded to each reviewer's notes, wherein we believe we've addressed all of the expressed feedback.

---

### Decision · Program_Chairs · 2022-01-20

**Decision:**

Accept (Poster)

**Comment:**

This paper show that in several different neural network  architectures, recurrent
networks that share parameters over iterations have comparable
performance and similar features to feed-forward networks of the same
"effective depth".

Reviewers initially had some reservations about novelty and
generalizability to deeper SOTA networks.  These were successfully
addressed by the authors and all reviewers feel the paper is above the
bar due to the importance of the area, and that this paper brings
together many important insights that, while many may have been known
before, had not previously been all brought together before.  The maze
task was also considered a useful task for the field.  I agree that
the paper makes a worthwhile contribution and am in favor of
acceptance.